# An Examination of US COVID-19 Vaccine Distribution in New Jersey, Pennsylvania, and New York

**DOI:** 10.3390/ijerph192315629

**Published:** 2022-11-24

**Authors:** Ethan Moss, Natasha A. Patterson, Brenda F. Seals

**Affiliations:** Department of Public Health, The College of New Jersey, Ewing, NJ 08628, USA

**Keywords:** pandemic, vaccination, vaccine, vaccine distribution, public health

## Abstract

COVID-19 quickly spread across the United States (US) while communications and policies at all government levels suffered from inconsistency, misinformation, and lack of coordination. In order to explain the discrepancy between availability and population uptake, a case study was conducted analyzing vaccine rollout plans, social media, and Health Officer/Other Key Informant interviews in New Jersey, New York, and Pennsylvania. Key research questions included, “What were the barriers and facilitators of early COVID vaccine distribution?” and “What mechanisms in the community emerged to alleviate strains in early vaccination?” Findings from this study revealed that pre-existing emergency preparedness infrastructures and plans developed since the 9/11 tragedy were seemingly abandoned. This caused health departments at all levels of government to make impromptu, non-uniform decisions leading to confusion, vaccine hesitancy, and ultimately low uptake. The results indicate that future vaccine rollout best practices must include evidence-based decision-making, coordinated communications, and outreach to high-priority and vulnerable communities.

## 1. Introduction

When this study concluded on 15 November 2021, over 185 million cases and 4 million COVID-19 related deaths occurred worldwide [1]. The United States (US) accounted for over 33.7 million cases and over 605,000 COVID-19 related deaths [1]. Despite the widespread availability of three effective US vaccines [2], only about 157,636,088 or 47.5% of the population were fully vaccinated [3]. As of 28 September 2022, the CDC reports that 67.9% of the US population has completed their COVID vaccine primary series, just short of the WHO recommended 70% target [4]. If roughly 5–10% of those remaining are unable to take the vaccine due to physiological complications (e.g., immunodeficiency disorders, being in current cancer treatment or others), 20% of the population remains COVID vaccine resistant.

Globally, systematic reviews estimate that intention to receive the COVID vaccination ranges from 27.7% to 93.3% [5,6,7]. These reviews note that hesitancy is higher among people who are women (especially those breastfeeding or pregnant), younger, lower income and education, and ethnic and racial minorities. Those who are hesitant have concerns about perceived risk, vaccine safety as well as questions about how the vaccines were developed and doubts about the approval process. Many perceived politics to have unduly influenced the vaccine development process [8]. Of particular note are concerns about vaccine hesitancy among health care professional and nurses, who voice concerns about vaccine safety and efficacy and echo a mistrust of health authorities [7]. The negative information about COVID in social media and having low confidence in the health care system were particularly influential on decisions not to vaccinate.

We conducted a literature review variously searching for COVID and “vaccine distribution”, “best practices” and “vaccine uptake.” We found only one study was based on local distribution experience. Washburn et al. [9], based on experience in Orange County, CA developed a checklist of best practices. The list focused on better use of vaccination data, allocation and distribution practices, reduced barriers for marginalized communities (e.g., addressing cost, technology needs, transportation and logistics) as well as leveraged the influence of trusted health care providers. They concluded by calling for a focus on community outreach and communication to foster transparency, inclusivity and trust in vaccines. Their list of recommendations is echoed in calls for solving problems of distribution, responding to challenges of primary care and problems of vaccine dissemination (e.g., setting up vaccination clinics and logistics, health care provider needs, and procurement and supply chains), and generally building public health infrastructures [10,11,12,13]. Beyond these pragmatics, most authors noted the critical need for better communication and leadership. “We certainly must avoid another situation where public health authorities and politicians are left to fly blind and then try to clean up the damage later [14].”

Reviews of health equity similarly had long lists of recommendations, the most inclusive being Bayati et al. [15]. In their review of 4623 articles, only 22 met criteria for studies of vaccine hesitancy of which 15 were conducted in the U.S. [15]. Notable changes need to focus on public health infrastructure and health systems (e.g., information systems that could reach diverse audiences, healthcare access and sufficient medical and medical-related facilities as well as equipment needed for storing and transporting vaccines). Politics that influence vaccination allocation rules must be addressed, as well as ideologies and policies that support racial bias and disenfranchise vulnerable populations.

Interventions that would address part of the aforementioned recommendations might be feasible, but very few actual trials are in place [16]. Of the studies of interventions, the most promising strategies to improve vaccine uptake were incentives to parents and health care workers, making information more salient and using trusted community members as messengers/advocates for vaccines [17]. Personalizing communications to the needs and concerns of communities and sending reminders may also be beneficial [18]. Unfortunately, across all of these studies, the level of analysis did not reach to the local level where “on the ground” decisions had to be made delivering vaccines. The long lists of recommendations remained at an abstract, policy or political level.

In order to better understand the discrepancy between vaccine availability for this important public health intervention and population uptake, a case study was conducted of vaccine rollout plans, related social media and key informant interviews for New Jersey, New York and Pennsylvania to identify successes and failures and provide a best practice framework for future emergency responses including mass vaccinations. This study sought to tell the story and context behind local vaccine distribution while detailing lessons learned at both the local and state policy levels. Our research questions centered on explaining “what happened” from a state/local health department perspective in an area highly impacted by COVID.

New Jersey, New York, and Pennsylvania, such as all states across the nation, struggled to find equitable, feasible, and timely approaches to delivering the newly approved COVID-19 vaccines. The distribution of vaccines was complicated by supply chain issues and political interference resulting in states having differing amounts of vaccine [19]. As a result, the federal plan quickly devolved to become state and city responsibility for vaccine distribution. Neighboring states, including the three states analyzed, lacked coordinated responses and ended up at conflicting phases which conflated messaging about availability, creating mass confusion [20]. In addition, New York City and Philadelphia had their own distribution plans, including unique phase definitions from that of their respective states. Vaccine rollout was less than ideal as, for example, states experimented with regional, mass versus local distribution, had underdeveloped registration websites and did not account for vulnerable populations that often lacked technology and software sophistication, such as the elderly. With community distrust mounting from the economic consequences of COVID-19 lockdown and other pandemic response measures as well as unequal mortality rates, some vulnerable groups and minorities, especially those with historic distrust, opted out of vaccination with historic rates of vaccine hesitancy.

The policy of the phased distribution was designed to create a system where the newly created COVID-19 vaccines could maximize benefits and minimize potential harms, promote justice, mitigate health inequalities and be a feasible and realistic approach to ending the COVID-19 pandemic through timely achievement of herd immunity [21]. As per the CDC and the Advisory Committee on Immunization Practices [2], the effective and equitable distribution of the vaccine was to be the primary responsibility of the federal government with the “flexibility within national guidelines [for] states and local jurisdictions to have flexibility to administer vaccine based on local epidemiology and demand” [22]. Under this theory, states would have the authority and ability to efficiently distribute the vaccine to at-risk populations, and marginalized communities and meet the specific needs of their residents. The gap between this policy and the history of vaccine distribution needs exploration, hence the purpose of this study.

The policy of the phased distribution was designed at the federal level [21] to create a system where the newly created COVID-19 vaccines could maximize benefits and minimize potential harms, promote justice, mitigate health inequalities and be a feasible and realistic approach to ending the COVID-19 pandemic through timely achievement of herd immunity. As per the CDC and the Advisory Committee on Immunization Practices [2], the effective and equitable distribution of the vaccine was to be the primary responsibility of the federal government with the “flexibility within national guidelines [for] states and local jurisdictions to have flexibility to administer vaccine based on local epidemiology and demand [22]”. Here, states would have the authority and ability to efficiently distribute the vaccine to at-risk populations, marginalized communities, and meet the specific needs of their residents.

For the COVID-19 pandemic, a phased approach was developed based on risk, essential services, and voluntary uptake in hopes that early distribution would meet the most urgent population needs and protect vulnerable populations in an equitable manner as follows: Phase 1A: healthcare workers, Phase 1B: essential workers/people at high risk/65+, Phase 2: critical populations, and Phase 3: general population [21]. Ideally, each phase would progress as vaccine availability increased and the target population vaccination rate peaked until widespread availability and uptake would achieve herd immunity.

## 2. Materials and Methods

### 2.1. Case Study Approach

We use a case study approach to generate a multi-faceted understanding of a complex, contemporary phenomenon [23]. An established design in the social sciences, case studies are mostly used in anthropology and ethnography. The strategy is to use multiple qualitative and quantitative data sources that allow the research team to triangulate analyses and interpretations to describe and explain an event in-depth. In this case, the unit of analysis is bound by time and place restrictions; specifically, the focus is on the early release and distribution of COVID vaccines in the Philadelphia/Central New Jersey area. While this design may not use the controls of other research designs that provide rigor, the case study provides face validity through transparency by presenting much of the data in their original forms so that the reader can review the “raw” data and make judgments and interpretations alongside of the researchers. Our research questions were, “What were the barriers and facilitators of early COVID vaccine distribution?” and “What mechanisms in the community emerged to alleviate strains in early vaccination?”

### 2.2. Government Website Review

Government health department websites for New Jersey, New York and Pennsylvania were reviewed. Health department websites for the cities of New York and Philadelphia were also reviewed because they functioned as their own entity during the distribution process. These cities received vaccine doses directly from the federal government and did not have to go through the states. They also had their own eligibility criteria separate and sometimes different from that of their state.

To document the differences between each Department of Health, a table was created to record the website content, including: “phase definitions,” “online tools,” such as apps and QR codes for appointments, “languages,” if the information was available in languages besides English, “computer literacy/accessibility,” to identify ways the public would be able to contact government staff for questions or to make appointments, “press conferences,” which were usually videos of daily or weekly press conferences held by government officials disseminating information about COVID-19 and the vaccine, and “sign language available” during the press conferences.

### 2.3. Key Informant Interviews and Survey

Key informant interviews were conducted with Public Health Officers involved in the COVID vaccine dissemination process from the Philadelphia metropolitan and Central New Jersey areas. Those recruited for this study worked at the municipal level. Similar to most government officials, they must meet licensure requirements (e.g., https://www.nj.gov/health/lh/professionals/licensing/, accessed on 1 September 2022). The Public Health Officers in this study are chief executive officers or upper management, and are responsible for overseeing the evaluation of health problems and planning and implementation of appropriate measures within their jurisdiction (municipal/regional/county/state). Local Health Officers plan and conduct emergency management and were primarily responsible for COVID response. Interview questions were developed based on the literature of best practices in vaccine dissemination and were pilot tested by practice interviews with a small sample of health officers who gave us feedback on question wording and organization. The finalized, open-ended questions focused on strengths, challenges, obstacles and accomplishments for vaccine distribution. Public Health Officers were identified via official websites and recruited. Most respondents were males between the ages of 25–55 years old. Sampling was augmented with a snowball technique where participants recommended colleagues and provided contact information. The confidential, one-time interviews were conducted by one researcher (E.M.) with an accompanying online survey.

Quantitative questions from the survey:

Questions 1–5: Demographic data

Question 6–8: In your opinion, on a scale of 0–10 rate how effective communication was concerning the COVID-19 vaccine and distribution, since January 2021. Asked separately for Federal, State and County Governments

Question 9: In your opinion, on a scale of 0–10 rate how effective the above 3 government agencies worked together to facilitate a speedy, safe, and equitable vaccine distribution plan since January 2021—Effectiveness of communication between federal, state and local governments (Extremely confusing, Slightly confusing, Extremely clear, Slightly clear)

Qualitative Interview style questions:

Question 1: How equitable do you believe the vaccine distribution process has been since January 2021? If not equitable, what populations were impacted and how?

Question 2: What has been the most significant challenge in the vaccination process since January 2021?

Question 3: How could the vaccination process be accomplished more efficiently?

### 2.4. Social Media Search

Each state and local health department used social media to disseminate information about COVID-19 and the vaccine. For the state and local health departments, direct links to their social media were provided on their official webpages. Many of them had their YouTube video press conferences embedded in the webpage. The social media content used in this study was based on viral posts. Official Facebook, YouTube, Twitter and Instagram pages were reviewed. Daily press conferences from each state streamed live on the social media sites. The comment section of the live stream posted to the social media sites were political in nature, including either positive or negative feedback on the political figure who was sharing the information in the video. More substantial vaccine information-seeking comments were found on local pharmacy and community social media sites.

The social media sites for small local pharmacies and vaccine hunter Facebook groups became very popular. They were able to utilize social media to make people aware that they had vaccines available and to recruit volunteers to work the vaccine distribution events. One pharmacy’s Facebook and Instagram post went viral because the pharmacist dressed up in a superhero costume while giving out the vaccine. He was interviewed by news outlets. Other similar pharmacy posts made traction with their amount of information they provided about COVID-19, the vaccine and having appointments available.

The most popular vaccine hunter groups were public, and anyone can join or view the posts. Different members of the group would post about available appointments, provide locations and times, asking if anyone wanted them, and several people would respond to the post. These posts were shared and reshared to the main Facebook timeline.

Once the pharmacy and vaccine hunter social media content were identified and addresses confirmed as being in NY, PA, or NJ area, we reviewed the content for 30 days.

## 3. Results

This study explores New Jersey, New York, and Pennsylvania health department responses in the midst of a pandemic, including news articles, government websites, social media, and interviews with Public Health Officials. Having large state-wide populations, New Jersey (8,882,190) New York (19,453,561) and Pennsylvania (12,801,989) also contain 2 major metropolitan areas, New York City (NYC) and Philadelphia, which are epicenters for viral spread [24]. The US government vaccine distribution plan considers NYC and Philadelphia’s Departments of Health separate entities from their states so that they receive their own vaccines and other supplies and have their own funding streams.

### 3.1. Government Websites

Since each state ultimately was responsible for their individual vaccine dissemination plan, a national standardized system was not developed and rather each state developed their own individual platform. New Jersey, New York, and Pennsylvania all utilized online tools to help facilitate what was perceived to be easy access to vaccines. New Jersey utilized the NJ Vaccine Scheduling System (NJ VSS), New York the “Am I eligible” app and Pennsylvania utilized a provider map on the state website. Each platform offered differing accessibility and alternate features, see Table 1. However, all platforms assume that the users have technology and software expertise.

New Jersey, New York and Pennsylvania had active State Health Department websites with up-to-date COVID-19 information on it, resources and locations for vaccination dissemination (Table 2). The vaccine sites were organized by county, city and zip code. New York City and Philadelphia had their own vaccine deliveries and did not have to rely on supply from the states. Both cities also had their own criteria for each phase of the vaccine rollout. New York State and New Jersey had a call center for those who had technological challenges. Both states also had a QR code for making appointments. Pennsylvania had a provider map so that the user could click on their own area to find a vaccine site to make their appointment. Most local jurisdictions were allocated vaccines from their state government. NYC, Philadelphia and select other major metropolitan cities across the US were allocated vaccine directly from the federal government. Differently timed and defined phased plans for each state and city then caused inequalities and general confusion between major metropolitan areas and the surrounding suburban neighborhoods, even within specific states and especially across state lines. For example, some cities and states identified the elderly population as 65 and older while others defined it as 75 and older.

The review of content and ease of use revealed New York City had the easiest navigation with respect to finding information. Figure 1 shows an easy-to-understand layout with information ranging from vaccination requirements, vaccination clinic locations, information on vaccines, COVID-19 testing, mental health resources and more all housed in one convenient easy to navigate site. When assessing for readability, all of the sites had a higher-than-average reading level for posted information. Some of the website’s information is available for language translation. New Jersey had English and Spanish. New York and Pennsylvania had multiple translations available. Leaders of each state and major city held daily press conferences. Sign language interpreters were present at each conference.

Figure 1 shows the NYC COVID-19 website on 09.09.21 and is representative sample of the ample information provided by NYC on the website. Information ranged from mental health resources, vaccine requirements, to pregnancy and COVID-19 information.

### 3.2. Key Informant Interviews and Survey

Responses from seven Health Officer Key Informant interviews and an online survey given to the same seven highlighted the challenges faced by public health professionals in adequately, equitably, and efficiently disseminating both vaccines and information about them. A limited number of key informants were contacted in order to obtain high-quality data from multiple sources quickly and efficiently. All but one of the seven respondents ranked the Federal and State governments’ communication below five (low effectiveness) and most ranked county communication below seven. Communication between the three levels of government was consistently ranked below five. Communication from all entities was also ranked either “extremely confusing” or “slightly confusing” by all respondents but one.

Highlighted quotes from the question on vaccine equity include:


*“Registration for vaccine interest was completed using the internet, almost exclusively, marginalized communities (i.e., people of color, the poor, the transient) may not have had the opportunity to register for their vaccine interest. We did not go door to door to identify people’s interests and register them (with acknowledgment of their limitations—no transportation, etc.).”*



*“Not equitable, Philadelphia missed the mark with partnering with diverse organizations to ensure equitable access and distribution of the vaccine to the most vulnerable populations”*



*“The Latino population was receiving mixed messages “Operation Warp Speed” was not clearly, explained…Very unreliable source President Trump contradicting the experts on national television…Materials not available in Spanish for low literacy levels…There did not appear to be great interest in providing, clear information to the Latinx population. Mega sites were usually not located in poor communities, limited hours, and roll out was inefficient for example in [name] county the Mega Site was open 3 days from 9am-3pm....Outrageous!”*


Regarding distribution challenges, respondents’ quote include:


*“Understanding the ever changing tiers” (phases)*



*“Vaccine delivery chain issues plagued the rollout”*



*“No rollout of public education from federal, state or local government…this generated mistrust and fear”*


Participants noted the following best practices:


*“Provide vaccines to local health departments that best know their community/populations. Local health departments providing vaccine/medication to the community in times of crises/emergency/pandemics has been the overarching plan for decades. COVID-19 vaccine deployment in New Jersey has not followed that model.”*



*“Follow the plans... They were there for a reason, but abandoned without explanation.”*



*“Public health efforts must start with public education. There must also be a plan for distribution that is equitable. Federal, state, and local governments have to work together.”*



*“the government must work with a diverse group of organizations that work within the community. Partnering with these organizations helps vaccinate the most vulnerable populations.”*


### 3.3. Vaccine Availability via Social Media

The social media pages for the state and county health departments as well as local pharmacies were also reviewed for information. The chain pharmacies such as Rite Aid, Walgreens, and CVS had limited appointments and once the appointments became available, they were already taken [25]. The demand for appointments was high. People often had to wait until the middle of the night to secure appointments as soon as the appointments were available. Many people on social media sites complained about having to refresh their screen several times before appointments became available. If the pharmacy was offering the Moderna or Pfizer vaccine, the first and second vaccine appointments had to be made together. This presented another challenge when there would be vaccine appointments available for the first shot but not the second shot therefore no appointment could be made. They would have to start all over for a new search or go to another pharmacy’s website.

Local, “mom & pop” or independent pharmacies began using social media to disseminate information about the vaccine and its availability. The smaller pharmacies partnered with community sites such as schools, colleges, and recreational centers to offer the vaccine. They also shared information about how independent pharmacies were not receiving as many vaccines as some of the larger pharmacies and mass vaccination sites. They recruited volunteers to help with the dissemination. Some drew much media attention as the community heroes filling in the gap for hard-to-reach communities. One pharmacy owner in PA dressed in a superman costume garnered much attention from many outside of the community. Dr. Ala Stanford, who created the Black Doctors COVID-19 Consortium (BDCC), also gained notoriety for her efforts in the community for first partnering with Black churches to provide COVID-19 testing to community residents who were left out of the process in Philadelphia, then shifting outreach efforts to acquire the same community residents vaccinated. During the Phase 1A and 1B, she hosted a Vaxxithon for 24 h and vaccinated over 5000 people including many over the age of 75 and those who were immunocompromised. She utilized Instagram and Facebook to let the community know where she would be on any given day [26]. She and the Black Doctors COVID-19 Consortium hosted vaccine events throughout the city of Philadelphia and Montgomery County, PA. The lines would be long and crowd control was needed. Oftentimes, there were issues with others who were not from the community traveling to these sites to acquire vaccines. Due to this, Dr. Stanford enforced a zip code policy, only offering vaccines to those who were from the most marginalized parts of the city with the highest rates of positive cases and deaths from COVID-19.

“The demand of vaccines is high, the supply of it is low, but you can bet that when others are sleeping, your team at xxx Pharmacy is doing whatever they can to get these vaccines into our hands so we can get them out to our community [27].” The social media sites for the local pharmacies grew with followers as they were able to deliver more vaccines. They became a lifeline for many who were desperate to become vaccinated but had challenges securing appointments.

“Vaccine hunters” began to emerge. These were people who had the time, the access to technology along with technological know-how to help members of the elderly community find or hunt down vaccine appointments. They would identify pharmacies that had appointments and make posts in the Facebook groups to see if anyone wanted the appointments. If they did, the vaccine hunter would complete the online appointment forms for the person. This was first seen as a benefit to those who did not have access to technology and time required to make vaccine appointments. Later, vaccine hunters were used to “cut the line” in front of those who were already marginalized and challenged in gaining access to the vaccine.

## 4. Discussion

### 4.1. Historical Lessons Learned and Community “Buy-In”

The SARS/MERS/H1N1/COVID viruses and their variants have long plagued the world. In addition, the most recent response was not the first. The US’s response to the 2009 H1N1, also known as the “swine flu”, was also a threat with a vaccination response. By mid-June WHO declared H1N1 a pandemic and by late April, a vaccine was already under development. The CDC over-promised availability by claiming 120+ million vaccines would be ready by December but delivered less than 17 million [28]. This limited supply of the H1N1 vaccine also resulted in a phased approach to vaccine the most needed populations first. This included healthcare workers, the elderly, and those with severe medical conditions leaving them more exposed to illnesses and death. “A Gallup survey from early November 2009 found that 54 percent of adults said the federal government was doing a poor or very poor job of providing the country with an adequate supply of the H1N1 vaccine [29]”. We identified only one study of vaccine intentions where willingness to receive a pandemic vaccine ranged from 8–67% [30]. Despite these drawbacks, the CDC estimates that 27% of those over 6 months old and 34% of high priority groups were vaccinated against H1N1 [28].

In the 11 years since the H1N1 vaccine rollout, protocols, policies, and plans were drawn up based on the success and failures of the H1N1 pandemic, but these procedures and plans were immediately disregarded by the federal government at the start of the COVID-19 pandemic to instead be replaced with *Operation Warpspeed* [28]. Likely leading the US to complete a scientific feat previously unheard of in terms of creating a vaccine so fast with a fairly rapid dissemination. Unfortunately, this *Operation* also led the US down a similar path of miscommunication, distrust, and an un-unified and often unjust vaccine distribution leading to inequities and vaccine hesitancy.

As of August 2022, we identified over 17 articles on COVID vaccine hesitancy including a wide range of example groups such as health care workers [31], those with a disease [32], older populations [33], children [34] as well as groups historically affected by structural racism where COVID only made inequalities worse including poorer health outcomes [35]. Yet in all of these articles, few recommendations beyond improving outreach methods were given. Many cited the need to establish trust and to address social determinants of health but beyond repeating what has become a longstanding “call” to address inequality, few concrete actions were given. Two examples focused on using community processes to better prioritize vaccine distribution and increase equity, but even here the authors seemingly had no documented accomplishments [36]. In the midst of a backlash against science, legal challenges to COVID vaccination regulations and policies may further threaten this public health intervention [37]. Without ongoing mechanisms to get communities on board with public health initiatives, best practices for vaccine distribution may be moot [38].

### 4.2. COVID-19 Vaccination Distribution and Logistics Challenges

Part of the success of the H1N1 vaccine distribution was that it worked with the Vaccines for Children Program, a CDC run program that allows states to acquire and administer vaccines for children who would otherwise go unvaccinated at no cost [29]. This program dates back to the late 1990s and has grown in popularity and success since then. However, a similar federally run program does not exist for adults [39], only state and some local/tribal groups can participate, and its distribution process is not designed for fast distribution [29]. The US should consider developing such a program for adult vaccinations and make both the VFC and an adult vaccination program amenable to quick distributions. As the VFC program is designed to administer already FDC-approved vaccines recommended by the Advisory Committee on Immunization Practices [2] it is positioned to be a model for pandemic response. Making better links between the VFC and municipal Departments of Health could facilitate both emergency planning and realistic exercises. Coupling such vaccine programs with community outreach, may improve adult and child vaccination rates for a broad spectrum of available vaccines [40,41].

Especially early in the pandemic, vaccine demand outpaced health department/health systems ability to secure and deliver shots. Due to the federal government’s lack of a unified distribution process for the COVID-19 vaccine, states, local health departments, and even private entities stepped up to play a major role in the vaccine distribution process [25]. Some of these companies included Shoprite, Wegmans and other big retailers. However, all vaccine providers faced a monumental logistical challenge: i.e., the need for deep refrigeration. The vaccines produced by Pfizer in New York City and BioNTech in Mainz, Germany required −70 °C (−94 °F) vs. the Moderna vaccine which required only −50 °C (5 °F) refrigeration [42]. Unused, opened vaccine vials had to be discarded introducing substantial waste of this precious commodity [43]. While meeting the needs of storing products such as vaccines may not be anticipated in emergency preparedness planning, working with industry for distribution, supply lines and storage and working with them for developed partnerships may expedite the success of future emergency management [44].

### 4.3. The Phased Approach

Since both vaccines became publicly available within days of each other, states had to choose between which vaccines to order and how much. They also needed to decide who would acquire the initial shipments, and how to best distribute them within their respective states (and major cities including NYC and Philadelphia). During the COVID-19 pandemic, a protocol similar to that of the H1N1 vaccine distribution was utilized which activated the VFC program to also allow states to procure EUA FDC COVID-19 vaccines and distribute them within their states (as well as some pre-decided local health departments including NYC and Philadelphia [29]). In policy and in theory, the phased approach allows for an equitable, timely and organized manner in disseminating vaccines to the public. In practice, the phased approach privileged first responders and those with good technological access, but left behind many vulnerable populations [45]. In addition, other forms of structural racism (hospital access and segregation, health department infrastructure by zip code, fears of accessing care based on immigrant/citizenship status, etc.) lead to unjust differential in mortality and morbidity [46]. If emergency preparedness is to provide equitable access to preventive services and care as well as serve all populations living in the US, planning must address ethical considerations for vulnerable populations.

### 4.4. Communication

Communication is a vital component of emergency response. During the COVID-19 pandemic response, communications were unorganized, confusing, misleading, and at times contradicting between all forms of government, medical associations and practitioners and communities ultimately undermining science-based decision-making. While the current administration published priorities committed to remediating such problems, without buy-in, practice and effective, coordinated planning, vital forms of communication that would facilitate an effective response may again go unused [47]. White House commitments to free testing, vaccination and treatment, personal protective equipment, and clear communication to communities remains “pie in the sky” promises without “on the ground” commitment requiring communication infrastructure and systems to implement noted strategies. Already, many Departments of Health are losing positions such as “outreach coordinators” and resources are being diverted away from building and rebuilding public health as relief funds, supplemental funding and assistance programs are called into question in political battles [28,48].

### 4.5. Accessibility

For the average community member and community health workers, eligibility for the vaccine was and is confusing due to a lack of meaningful communication from Federal, state, local, and private (CVS, Rite Aid) distribution entities who failed to account for language barriers, education level of residents, and general miscommunication. Access to the vaccine was limited based on social determinants of health: transportation, technological accessibility, language barriers, and general health education failures.

Transportation to the vaccine sites was an unaccounted-for factor during the beginning phases of the vaccine rollout. Many vaccine clinics, especially initially, were larger sites where people had to travel farther from their homes. For those without access to a car or the inability to drive, limited access to public transportation, for funds to pay for public transportation, this hindered their ability to reach those sites since vaccines were not widely available locally. More so, most vaccine clinics were held during traditional working hours thus giving many people the choice between going to work or acquiring a vaccine. This issue was further exacerbated by the likely side effects of the vaccines which made missing a day or two of work highly likely and then needing a repeat dose in a few weeks. Furthermore, finding a vaccine appointment in general was very difficult and sometimes required checking websites at off peak hours such as in the middle of the night.

A lack of health education materials and general misinformation spread on social media combined with language barriers to non-English speaking residents also caused confusion and mistrust in many communities. Not all communities have equal access to education which results in lower health literacy rates and thus a harder time understanding the very complex messages released by the CDC and other agencies. More so, computer access and computer literacy also played a role in limiting vaccine accessibility to some populations as most communication and clarifications of press releases was carried out online.

### 4.6. Policy Implications and Framework for the Future

A best practices framework for efficient and equitable vaccine distribution during a public health emergency including pandemic and epidemic responses would include 3 pillars (Figure 2). One, effective leadership and science-based decision making. Two, effective consistent communication. Three, laying the groundwork for reaching high priority and vulnerable communities. For the last, many in public health argue for improving public health communication infrastructure to build an ongoing relationship with communities. This relationship is key to increasing knowledge of and trust in public health measures. This system would need consistent monitoring and flexibility to strategically respond to crises while prioritizing equity. Each pillar could then be expanded upon and modified for any public health emergency. Furthermore, emphasis on the connection between local health departments and the communities they serve will play an essential role in successful response.

In the case of COVID-19, pillar one must be the foundation of federal government responses based on science and evidence to categorize sub-populations based upon risk, and follow protocols to modify existing infrastructure and communication plans as the emergency response develops over time. Pillar two points to the necessity for internal communication between all levels of government and their partners in the scientific and logistics communities prior to pre-arranged press-releases of information for consistency across geographic areas. Pillar three would ensure that public health measures similar to COVID-19 vaccination distribution sites would be easily accessible to those in low-income communities, with community leaders who could quickly serve as language interpreters, and distribute information on the pandemic. The latter would make the response resilient for and accessible to low literacy communities, the elderly and immigrant populations for hundreds of cultural and linguistic groups. These three pillars are the foundation for effective pandemic/epidemic responses that would streamline the public health emergency response infrastructure and increase trust in the government and scientific community.

## 5. Limitations to the Study

This overview case study focused on communications and key informants from the NY, NJ, and PA areas primarily around Philadelphia. The main limit of a case study design is that, because the results are time and place bound, generalization may be questionable. We focused our recommendations on policy in order to make the findings of this study widely applicable. Due to public health officers being immersed in the vaccine distribution process and addressing pandemic related needs and challenges, many found it difficult to participate which limited the sample. The limited sample size also made it impossible to know if the data have been saturated [49]. While the social media and websites were all available, our sample reflects the response in the Northeast US and may not be generalizable. Many of the comments in the discussion section reflect our experience aiding in COVID response to local health departments and may also be limited in scope. This unfunded study limited the resources and opportunities to do the additional outreach needed to provide a more representative sample. Future research may want to investigate responses from a wider range of health departments and states and provide a more detailed analysis. Our goal was to provide the context and responses that would guide future planning. However, we were unable to identify any other sources for “ground up” information on the epidemic response which this paper uniquely addresses.

## 6. Conclusions

Our final literature search in March 2022 netted no articles on “COVID” and “vaccine distribution”, “vaccine” and “health department” detailing “on the ground” lessons learned about vaccine dissemination. This study sought to tell the story and context behind local vaccine distribution while detailing lessons learned at both the local and policy levels. Our Key Informants complained that much of the work based on emergency preparedness Point of Dispensure (POD) and H1N1 was ignored in the US COVID-19 response underscoring the need for public health leaders to create standardized Emergency Action Plans (EAPs) for mass vaccinations that coordinates political leaders, community leaders and relevant industry across all states. Such plans must be resilient to pandemics that could last years, sensitive to an early response where testing is key, a middle response where new vaccines are coming out, but supply chain and equitable distribution are challenging, and a later response when vaccines are plentiful but community uptake is needed to attain herd immunity. Determining quantifiable ways of defining at-risk populations and providing clear steps of how to vaccinate each member of each population is an important step in ensuring equitable distribution.

Vaccine hesitancy due to fear of side effects cannot be ignored. Mistrust in pharmaceutical companies in their production of vaccines has been a reason of distrust for many people, including healthcare workers [50]. With the rapid spread of COVID-19 across the world, the urgency to have a vaccine ready and available for dissemination may have exacerbated those fears. The general public may have been unaware of the new technologies and advancements in vaccinology. “These advances allow improved precision in vaccine design and more rapid manufacturing timelines” [51]. In addition to what seemed to be a quick process, the fear of side effects impacted the decision whether or not to be vaccinated. Albeit rare, reports of orofacial side effects and even death in any close proximity to the vaccine added to those fears [52,53]. Equitable mechanisms of communication that promote public health education must be prioritized to reduce vaccine hesitancy [54]. A transparent plan of how all population subgroups can and will receive vaccination must be consistent across all mediums and sectors. In addition, some of the heroes who arose overcoming challenges must be recognized including grassroots organizations and independent pharmacies. A wholistic community response is needed including government resources and support for social services, measures to increase safety for vulnerable populations and for unanticipated obstacles such as breaks in supply chains. The government must provide for a strong, ongoing public health presence at the community level in order to increase trust, practice and refine emergency response planning and increase community knowledge of science and public health practices to prevent, mitigate and effectively respond to pandemics.

## Figures and Tables

**Figure 1 ijerph-19-15629-f001:**
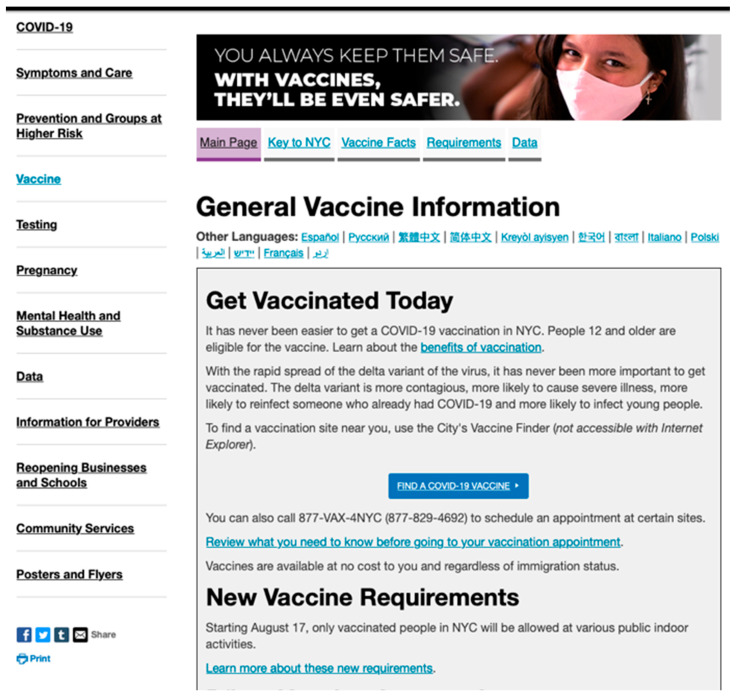
Snapshot from NYC COVID-19 Website.

**Figure 2 ijerph-19-15629-f002:**
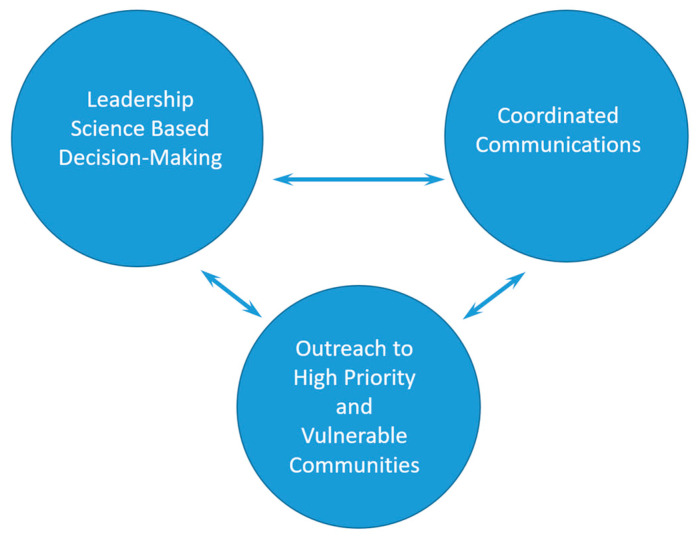
Best Practices for Mass Vaccination.

**Table 1 ijerph-19-15629-t001:** NJ NY PA COVID-19 Vaccines Rollout Comparison: Online Tools.

	Online Tools	Languages?	Call/Email For Appt for Computer Literacy/Accessibility	Sign Language Available for the Press Conference
New Jersey	NJ VSS QR Code needed	English Spanish	Call Center	Yes
New York	“Am I eligible” app and State form/waitlist QR Code used	Translations available	Call Center	Yes
NYC	“Am I eligible” app and State form/waitlist QR Code used	Translations available	Call Center	Unknown
Pennsylvania	Provider map	Translations available	Not seen	Yes
Philadelphia	Provider map	Translations available	Not seen	Yes

**Table 2 ijerph-19-15629-t002:** NJ NY PA COVID-19 Vaccines Rollout Comparison: Government Website.

	Phase Definitions February/March 2021	Phase Definitions April/May 2021	Phase Definitions June/July 2021	Phase Definitions August/September 2021	Phase Definitions Current Live Website
New Jersey	https://web.archive.org/web/20210223135149/https:/covid19.nj.gov/faqs/nj-information/slowing-the-spread/who-is-eligible-for-vaccination-in-new-jersey-who-is-included-in-the-vaccination-phases	https://web.archive.org/web/20210414114058/https:/covid19.nj.gov/faqs/nj-information/slowing-the-spread/who-is-eligible-for-vaccination-in-new-jersey-who-is-included-in-the-vaccination-phases	https://web.archive.org/web/20210720145801/https:/covid19.nj.gov/faqs/nj-information/slowing-the-spread/who-is-eligible-for-a-covid-19-vaccine-in-new-jersey	https://web.archive.org/web/20210822020246/https:/covid19.nj.gov/faqs/nj-information/slowing-the-spread/who-is-eligible-for-a-covid-19-vaccine-in-new-jersey	https://covid19.nj.gov/pages/vaccine
New York	https://web.archive.org/web/20210209195624/https:/covid19vaccine.health.ny.gov/phased-distribution-vaccine	https://web.archive.org/web/20210427112019/https:/covid19vaccine.health.ny.gov/distribution-vaccine	https://web.archive.org/web/20210710141800/https:/covid19vaccine.health.ny.gov/distribution-vaccine	https://web.archive.org/web/20220903051909/https:/covid19vaccine.health.ny.gov/distribution-vaccine	https://covid19vaccine.health.ny.gov/distribution-vaccine
NYC	https://web.archive.org/web/20210202143633/https:/www1.nyc.gov/site/doh/covid/covid-19-vaccines.page	https://web.archive.org/web/20210421123127/https:/www1.nyc.gov/site/doh/covid/covid-19-vaccines.page	https://web.archive.org/web/20210623035906/https:/www1.nyc.gov/site/doh/covid/covid-19-vaccines.page	https://web.archive.org/web/20210909161339/https:/www1.nyc.gov/site/doh/covid/covid-19-vaccines.page	https://www1.nyc.gov/site/doh/covid/covid-19-vaccines.page
Pennsylvania	https://web.archive.org/web/20210222141148/https:/www.health.pa.gov/topics/disease/coronavirus/Vaccine/Pages/Vaccine.aspx#distribution	https://web.archive.org/web/20210526211914/https:/www.health.pa.gov/topics/disease/coronavirus/Vaccine/Pages/Vaccine.aspx	https://web.archive.org/web/20210705054901/https:/www.health.pa.gov/topics/disease/coronavirus/Vaccine/Pages/Vaccine.aspx	https://web.archive.org/web/20210902141619/https:/www.health.pa.gov/topics/disease/coronavirus/Vaccine/Pages/Vaccine.aspx	https://www.health.pa.gov/topics/disease/coronavirus/Vaccine/Pages/Vaccine.aspx
Philadelphia	https://web.archive.org/web/20210203034834/https:/www.phila.gov/2021-01-12-city-announces-schedule-of-priority-populations-for-covid-19-vaccine/	https://web.archive.org/web/20210519232526/https:/www.phila.gov/2021-01-12-city-announces-schedule-of-priority-populations-for-covid-19-vaccine/	https://web.archive.org/web/20210730002638/https:/www.phila.gov/programs/coronavirus-disease-2019-covid-19/vaccines/vaccine-distribution/ **	https://web.archive.org/web/20210907074732/https:/www.phila.gov/programs/coronavirus-disease-2019-covid-19/vaccines/vaccine-distribution/	https://www.phila.gov/programs/coronavirus-disease-2019-covid-19/vaccines/vaccine-distribution/

** Note change in Gov Website address.

## Data Availability

Key informant data was originally collected by the authors and is not available due to confidentiality considerations as specified in the participant consent forms. All other data is publicly available via the internet, see specific citations in the references.

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
