# Peer review of "An Examination of US COVID-19 Vaccine Distribution in New Jersey, Pennsylvania, and New York"

_ijerph, 2022, doi:10.3390/ijerph192315629_

Round 1
Reviewer 1 Report
The paper is well done and it is clear, with many innovative ideas. The methods are adequately presented. I suggest that the authors change the introduction and discussion. In the introduction the authors should better clarify the purpose of the study because it is not well explained. In the discussion they should include a paragraph on fear of vaccines due to side effects. It is suggested to cite the following studies:
doi: 10.1001 / jamahealthforum.2021.0804
https://doi.org/10.1080/09273948.2021.1976221
10.3390 / jcm10245876
doi: 10.1001 / jamanetworkopen.2021.40364
10.3324 / hematol.2021.279075
https://doi.org/10.1111/jop.13165
Author Response
We would like to thank the reviewers for taking the necessary time and effort to review the manuscript. We sincerely appreciate all of your valuable comments and suggestions, which helped us in improving the quality of the manuscript.
The introduction and discussion were revised as suggested. The introduction now includes a review of best practices and interventions to facilitate vaccine distribution/uptake. For the discussion, a new paragraph focused on the fear of vaccines due to side effects based on four new studies including the studies provided in your comments. Thank you for the selected studies that we used to strengthen the discussion.
Reviewer 2 Report
I would first like to thank you for the opportunity to review this manuscript.
The paper captures and details, as the authors indicate, a local response to vaccine delivery through a case study.
I believe that the description of the methodology needs to be improved.
In the Material and Methods section I would dedicate an initial space to describe these types of studies.
It should better define the hypothesis, the objectives of the study and relate the questions to these objectives to make it easier for the reader to understand the object of the study.
I would incorporate a chronological description that better details the order of each of the phases.
In the results, describe better the number of informants interviewed in order to have a quantitative idea of the representation of the sample or if there is saturation of the discourse in the answers.
In the discussion, incorporate other successful international distribution and acceptance strategies and include possible interpretation and comparison in the analysis. Or organisation of the health system, accessibility, or tradition in the vaccination process.
Expanding the literature
I hope you will take these considerations into account to improve the quality of the article.
Thank you
Author Response
The literature review was expanded to include studies of vaccine distribution best practices and vaccine hesitancy. As suggested, in the Materials and Methods section, a paragraph on case study approaches was added along with supporting references. The limits of such a study were later discussed in the limitations section. We incorporated research questions both here and in the abstract to make it clearer.
The number of key informants was made more apparent. In the results, a section on why the sample size of key informant interviews was added. More so, in the limits of the study, a section commenting on how due to the small sample size we can not determine saturation was also added. This case study depended on a convenience sample of Key Informants. We note the limits of this approach in the discussion. We also expanded the discussion to include international references to be more inclusive of best practices. Again thank you for helping us improve the quality of this paper.
Reviewer 3 Report
I have carefully read through this manuscript and find some critical problems as below.
(1) A thorough review of literature is abesent in the introduction section, and readers almost know nothing about what has been done in previous relevant investigations, what research gap this study would like to fill, and what citation or reliable source could support the arguement and data as displayed by authors. For example, It is shown in line 29-30
| "only about 157,636,088 or 47.5% of the population were fully vaccinated" |
and unfortunately, there is no citation to support such arguement.
(2) What's the research contribution of this study? Readers could not find any statement about it.
(3) The reference format is inconsistent and messy in the main text and that at the end of the manuscript. The citation in the main text seems like the APA style, while that at the end of the manuscript seems like the AMA style.
Some reference such as
| Benjamin-Chung J, Reingold A. (2021) Measuring the Success of the US COVID-19 Vaccine Campaign–It’s Time to Invest in and strengthen immunization innocation system |
could not be found in the main text.
Author Response
All references and internal citations have been updated in both the narrative and the reference section to match the requirements of the journal, unused references were omitted, and new references were added accordingly. The literature review section was updated to include a review of vaccine distribution best practices and vaccine hesitancy. The methods section was updated to better describe the type of study and includes our research questions. The revised submission was reviewed carefully to only include references for articles cited in the main body.
Thank you for your thoughtful comments that helped us improve the paper.
Reviewer 4 Report
1. The abstract should be briefer. Please make it more concise and logical. In addition, the abstract should tell the reader precisely and briefly the results or conclusions of the study.
2. The introduction does not provide sufficient background and innovations or contributions of the paper.
3. The paper's policy recommendations are not very intuitively presented. I suggest to devote a special section to “Policy Implications”.
4. There are not enough graphs of data types in the paper.
Author Response
The abstract of the paper was re-written to be more concise and logical as suggested. The revised abstract also now includes key research questions and a summary of the findings/policy implications.
The introduction was expanded to address best practices in vaccine distribution and vaccine hesitancy. The methods section includes more information on the type of study as well as our research questions. We hope this details the contribution of our study.
The policy section was added to the discussion as well as expanding on implications and limits of the study.
We relabeled section 4.6 to read Policy Implications as we put this information in this section.
Because there was limited quantitative data, we chose not to present our data in graphic form. The case study approach highlights transparency, defined as presenting as much “raw” data as possible so that the reader can see the face validity of the data. To meet this request, we added a screenshot of the NYC COVID-19 website to better show readers visually truly how easy this site was for people to navigate and the convenience of readily available information.
Thank you for your delineation of ways to improve our paper.
Round 2
Reviewer 1 Report
The manuscript has been sufficiently improved and is ready for publication in my opinion. A few small changes in shape:
Line 542: the references should be placed together in the same bracket [52-53]. Finally, before some references the space before the parenthesis should be inserted, for example: "fully vaccinated [3]" and not "fully vaccinated [3]", as in other cases.
Author Response
THANK you for the kind edits. We have put in these corrections.
Reviewer 3 Report
While there are some improvements than the last version, there remain many problems to be addressed before the manuscript can reach the standard of publication.
(1) More details about respondents in the interview should be provided (e.g., demographics, political stance), which should be organized in a table.
(2) More details about the process of interview should be clearly stated to increase the credibility and avoid being trapped in the talk in generalization. Besides, the contents presented by interviewees should be shown in italic, following by the analysis of authors their own.
(3) The interview contents had better to be presented in a form of theme analysis where applicable if the contents of interviewees could be summerized into several themes and sub-themes. The current form of presentation still seems scattered.
(4) The presentation of table 2 is extremely unfriendly to readers. Authors should consider how to display and organize the contents in details, and avoid just simply placing hyperlinks in the main text.
(5) More details about how the contents of social media are selected and analyzed should be provided and clearly stated. The current presentation lacks necessary information.
Author Response
Thank you for your thoughtful review and suggestions.
(1) More details about respondents in the interview should be provided (e.g., demographics, political stance), which should be organized in a table.
We added a better description of role of the Health Officer, age and sex into the narrative. Because of the small number of Health Officers in the recruitment area, we refrained from putting this information in a table (see our rationale below under revision (3). Please note that we did not ask about political stance and cannot respond to that but instead provided more information on the occupation of Health Officer.
(2) More details about the process of interview should be clearly stated to increase the credibility and avoid being trapped in the talk in generalization. Besides, the contents presented by interviewees should be shown in italic, followed by the analysis of authors of their own.
We added a few sentences about the interview process and put all quotes in italics.
(3) The interview contents had better to be presented in a form of theme analysis where applicable if the contents of interviewees could be summarized into several themes and sub-themes. The current form of presentation still seems scattered.
Because health officers in the Philadelphia/Central New Jersey area are a small sample, presenting the demographics for each participant would allow them to be identified. We considered grouping the information but this violated the CDC/APA recommendation for protecting participants by having no cell sizes less than 10. We did add some information for age/sex.
While we experimented with regrouping our qualitative data from the interviews, the themes identified parallelled our original analysis which presents the quotes organized by the questions that we asked. Hence, we kept the original organization as this reflects the answers given by the participants to the questions asked and is truer to the linguistic responses of the respondents.
(4) The presentation of table 2 is extremely unfriendly to readers. Authors should consider how to display and organize the contents in detail, and avoid just simply placing hyperlinks in the main text.
We revisited changing this table and did not find a way to make it more user-friendly. This table is really a resource for readers to review and see for themselves. We struggled with how to better describe the visuals and unless we actually put in pictures and then pointed out aspects of the websites, this suggestion would not work. We thought this reoriented the paper and would like to leave this table as is.
(5) More details about how the contents of social media are selected and analyzed should be provided and clearly stated. The current presentation lacks necessary information.
We added more details about how the social media content was selected and analyzed to the methods section 2.4 Social Media Search.